# A Watt-Level Pulsed Er:Lu_2_O_3_ Laser Based on a TiB_2_ Saturable Absorber

**DOI:** 10.3390/nano14040379

**Published:** 2024-02-18

**Authors:** Yangyang Liang, Lu Zhang

**Affiliations:** 1China Key Laboratory of Laser & Infrared System (Shandong University), Ministry of Education, Qingdao 266237, China; liangyang@sdu.edu.cn; 2School of Information Science and Engineering, Shandong University, Qingdao 266237, China

**Keywords:** TiB_2_ nanoparticles, Er:Lu_2_O_3_ laser, passively Q-switched laser, mid-infrared laser

## Abstract

TiB_2_ nanoparticles with a bandgap of 0 eV were prepared, and the corresponding nonlinear optical response at 2.85 μm was investigated. Employing a TiB_2_ as a saturable absorber, a 2.85 μm pulsed Er:Lu_2_O_3_ crystal laser with an average output power of 1.2 W was achieved under a maximum pump power of 9.51 W. Laser pulses with durations of ~203 ns were delivered at a repetition rate of 154 kHz, which corresponds to a pulse energy of ~7.8 µJ and a peak power of 39.3 W. As far as we know, the result represents the highest average output power from all Q-switched Er:Lu_2_O_3_ crystal lasers.

## 1. Introduction

Solid-state pulsed laser systems operating in the mid-infrared range have demonstrated excellent performance in delivering high-energy laser pulses. Thus, they are attracting more attention in surgery, manufacture, atmospheric monitoring, and infrared countermeasures. Additionally, the laser can also serve as a pump source to drive optical parametric processes, extending the laser wavelength to a longer band [1,2,3].

The first laser at ~3 μm was realized with an Er^3+^-doped medium in 1967, which generated the stimulated transition from the energy level of ^4^I_11/2_ to ^4^I_13/2_ [4]. Nowadays, many Er^3+^-doped crystals (Er:YAG, Er:YLF, Er:Lu_2_O_3_, and Er:YAP) and ceramics (Er:Lu_2_O_3_ and Er:Y_2_O_3_) have been widely used to realize 3 μm lasers [5,6,7,8]. The Er:Lu_2_O_3_ crystal stands out among its counterparts due to its superior thermal and laser properties. Due to the similar mass between Lu^3+^ and Er^3+^, the latter distortion of Er:Lu_2_O_3_ crystals is smaller, and the thermal conductivity decay due to the doping is slighter [9]. Meanwhile, Er:Lu_2_O_3_ crystals with any doping concentration (0% to 100%) can be grown [9]. The heavy doping concentration will enhance the energy upconversion process of Er^3+^ ions; as a result, the laser can break the self-termination and even surpass the Stokes limit. Moreover, the low phonon energy of the crystal will reduce the possibility of non-radiative transmission of Er^3+^ ions, which leads to a higher fluorescence efficiency [10,11].

Pulsed lasers can be realized by employing modulators in the laser cavity. As novel optical modulators, low-dimensional materials feature properties of broad response bands, fast optical response, and ease of integration. Many outstanding pulsed lasers in the 3 μm region have also been reported; for example, a pulsed laser with a duration as short as 50 ns was realized with ZrC [12], and with a laser power as high as 1.13 W was realized with Ti_3_C_2_T_x_ nanoparticles [13]. However, compared with the output power reported in the continuous wave (CW) laser (>14 W) [14], a higher output power of the Q-switched Er:Lu_2_O_3_ crystal laser is worth being expected by optimized SAs.

Titanium diboride (TiB_2_) is a metal diboride with a hexagonal structure (space group P6/mmm). Recent studies show that the low-dimensional counterparts also maintain the hexagonal atomic structure and chemical composition [15,16,17]. Thus, the material exhibits high stability, high melting temperature, hydrolysis resistance, antioxidant capacity, and high LDT. Employed as an SA in a fiber laser, the TiB_2_ nanosheet has delivered ultrafast pulses with a pulse duration of 596 fs and a peak power of ~2 kW at emitting at 1559 nm [18]. The above property and result show that a mid-infrared pulsed laser with watt-level output power is worth being expected with TiB_2_. However, the optical response properties of the TiB_2_ nanoparticles in the mid-infrared region have been less reported.

In this paper, the band structure of TiB_2_ nanoparticles was calculated theoretically, which predicted the saturable absorption property of the TiB_2_ nanoparticles in the mid-IR region. The TiB_2_ nanoparticles were prepared using the liquid phase exfoliation method. We experimentally characterized its morphology and nonlinear optical response at 2.85 μm. The experimental results show the prepared TiB_2_ nanoparticles can function as SAs at 2.85 μm. The laser modulation ability of the TiB_2_ nanoparticles was investigated by integrating TiB_2_ SA into the Er:Lu_2_O_3_ crystal laser. As a result, laser pulses with a duration of ~203 ns and pulse energy of ~7.8 µJ were delivered at a repetition rate of 154 kHz, which corresponds to an average output power of 1.2 W and a peak power of 39.3 W.

## 2. The Electronic Property of TiB_2_ Nanoparticles

Figure 1a,b show a 2 × 2 × 1 TiB_2_ supercell used to explore the electronic properties of TiB_2_ nanomaterials. The geometry optimization and the band structure calculation of the supercell were performed with the Perdew–Burke–Ernzerhof (PBE) function of Generalized gradients approximation (GGA). Energy, maximum force, and maximum displacement convergences were set as 2 × 10^−5^ Ha, 0.004 Ha·Å^−1^, and 0.005 Å, respectively. The calculated band structure and density of state (DOS) are shown in Figure 1c,d, respectively. The calculated bandgap of the materials is 0 eV, which meets well with the result reported in Ref. [15]. The graphene-like band structure of the materials allows it to absorb any incidence photon [19]. As a tiny sample possesses a certain number of electrons, when high-brightness light incident on the materials, electrons will be gradually excited to a high energy level until the sample is bleached. In macroscopic terms, the material will show a saturable absorption property. Therefore, the saturable absorption property of TiB_2_, at ~3 μm, is also worth expecting.

## 3. The Preparation, Morphology, and Optical Response of TiB_2_ Nanoparticles

The TiB_2_ nanoparticles were prepared using the liquid phase exfoliation method, which features excellent controllability and low preparation costs. To start, the commercially available TiB_2_ powder was mixed with pure isopropyl alcohol (IPA) at a concentration of 1 mg/mL. The mixture was then sonicated for four hours to exfoliate the TiB_2_ nanoparticles from the bulk counterpart before being centrifugation at 10,000 r/min for 15 min to retain the nanoparticles in the supernatant. Afterward, the supernatant was gathered for further characterization and experiments.

The morphology of the nanostructured TiB_2_ was thoroughly analyzed. Figure 2a depicts the SEM image, which clearly shows the granular structure of the sample consisting of numerous small exfoliations with a lateral size of a few micrometers. Figure 2b displays the TEM image; the stepwise variation in the grayscale reveals that TiB_2_ samples are assembled by the layer component with a width of a few micrometers. The grayscale variation in Figure 2b corresponds well with the SEM image that the TiB_2_ nanoparticles are prepared. Additionally, the image indicates that some ultra-thin TiB_2_ samples were also prepared.

The accurate thickness information of the exfoliations is characterized by atom force microscopy (AFM). The AFM image and the corresponding thickness distribution of the TiB_2_ nanoparticles are shown in Figure 3a,b, respectively. The step-varied thickness curve A in Figure 3b states that the prepared nanoparticles are assembled by many layers. Both curves in Figure 3b demonstrate that the peak thickness of the exfoliation is as thick as 80 nm. The above morphology characterizations state that the TiB_2_ nanoparticles have been successfully prepared.

In Figure 4a, the Raman spectra of bulk TiB_2_ (top) and TiB_2_ nanoparticles (bottom) are shown. The Raman peaks of the TiB2 nanoparticles have shifted compared to the bulk samples. Specifically, the peaks have shifted from 261 to 255 cm^−1^, 261 to 255 cm^−1^, and 261 to 255 cm^−1^, corresponding to the vibration modes of B_1g_, E_g_, and A_1g_. If we ignore any measurement errors, the results match well with the findings of previous publications [20,21]. 

The optical response of the nanoparticles was characterized. The collected supernatant was spin-coated on a dual-end polished YAG substrate and dried at 80 °C for 10 min. The linearly optical transmittance is measured with an ultraviolet–visible near-infrared spectrophotometer (Cary 5000 UV-Vis-NIR, Agilent Technologies, Inc., Santa Clara, CA, USA). As depicted in Figure 4b, YAG-based TiB_2_ nanoparticles exhibit a smooth transmittance curve around an absolute transmittance of 80%, in the range from 1000 nm to 3300 nm. With a self-made Q-switched laser emitting at 2.85 μm, a clear saturable absorption property of the TiB_2_ nanoparticles was obtained and shown in the insert of Figure 4b, which corresponds to a saturable intensity of 0.46 mJ/cm^2^, a modulation depth of 3.7% and a saturable transmittance of 83.6%. The result shows that the TiB_2_ can be utilized as an SA and supports a pulsed laser operation.

## 4. Laser Experimental Setup

Figure 5 demonstrates the experimental setup of the TiB_2_ Q-switched Er:Lu_2_O_3_ crystal laser. In the setup, a fiber-coupled laser diode (LD) emitting at 976 nm was used as the pump source. The coupling fiber has a core diameter of 105 μm and a numerical aperture(NA) of t 0.22. The pump beam is focused on the gain medium via a 1:3 lens group (L1 and L2). The input coupler (IC) is a plane-concave mirror with a curvature radius of 10,000 mm, which is coated with both anti-reflectivity film (AR @ 950 to 1000 nm) and high-reflection film (R > 99% @ 2700 to 2900 nm). Three flat output couplers (OCs) with transmittance of 1%, 3%, and 5% at 2.85 μm were used to optimize the laser performance. The gain medium is an uncoated Er:Lu_2_O_3_ (7at.%) crystal with a dimension of 10 × 3 × 3 mm^3^, which is grown with the heat-exchanger (HEM) method [22,23,24]. The crystal is wrapped with indium foil and mounted in a copper block, which is cooled at 15 °C using a water-cooler. A prepared TiB_2_ SA is placed near the OC. The laser is separated from the residual pump via a dichroscope mirror (DM).

Output power is measured by a powerhead (S314C, Thorlabs Inc., Newton, NJ, USA) and a power meter (APM100D, Thorlabs Inc.). Laser pulse signals are detected using a fast HgCdTe IR photon detector (PVI-4TE-4, Vigo System, Ozarow Mazowiecki, Poland) with a response time of 1 ns. The signal was then observed on a digital phosphor oscilloscope (DPO 7104C, Tektronix Inc., Beaverton, USA) that has a bandwidth of 1 GHz and a sampling rate of 20 GS/s. Additionally, a Fourier transform optical spectrum analyzer (OSA 207C, Thorlabs Inc.) is used to detect the lasing spectra.

## 5. Results and Discussion

Initially, CW lasers were realized. As shown in Figure 6a, all the output powers increase linearly with the absorbed pump power. The three highest output powers of 2.23 W, 2.64 W, and 2.45 W were delivered with transmittances of OCs of 1%, 3%, and 5%. The error bars depicted in the figure represent uncertainties that are below 2%, which corresponds to slope efficiencies of 18.3%, 22.7%, and 20.5%, respectively. The laser spectrum at the output power of 2.64 W is shown in the inset of Figure 6a, which shows the central wavelength of the laser is 2.85 μm.

After integrating the TiB_2_ SA as a Q-switcher in the Er:Lu_2_O_3_ crystal laser, 3 μm laser pulses are delivered. Figure 6b depicts the properties of the average output power versus the absorbed pump power of the Q-switched lasers with different OCs. The highest average output powers in all cases are in the order of watts. With the 3% OC, the highest average output power of 1.2 W is realized, which corresponds to a slope efficiency of 15.0%. The maximum uncertainty of the output power of 5% for the Q-switched lasers is higher than that of the CW lasers. The corresponding lasing spectrum with a central wavelength of 2.85 μm is shown in the inset of Figure 6b. With the 1% OC and the 5% OC, average output powers of 0.96 W and 1.05 W are realized, respectively, corresponding to slope efficiencies of 10.8% and 13.3%. These results demonstrate that TiB_2_ nanoparticles are an excellent Q-switcher in delivering high-power laser in the mid-infrared band.

The evolutions of repetition rates and pulse durations with respect to the absorbed pump power are shown in Figure 7. The repetition rates increase as the absorbed pump power increases, which corresponds to an uncertainty of 8%. The maximum pulse repetition rate of 166 kHz is achieved in the T = 1% Q-switched laser system. Figure 7b shows that the pulse duration is reduced when the absorbed pump power increases. The shortest pulse duration of 203 ns is realized with the OC of T = 3%, whose uncertainty is 5%. In Figure 8, the output pulse train and the single pulse shape at the highest repetition rate are presented, which indicates the output laser pulse is relatively stable. The maximum amplitude difference (peak to peak) is estimated to be less than 10%. It is noteworthy that the stable pulse train can also be regenerated from the TiB2 SA-based Er:Lu_2_O_3_ laser when the TiB2 SA has been exposed to free space for a month. This demonstrates the excellent hydrolysis resistance and outstanding antioxidant capacity of the TiB2 SA.

Figure 9a,b present the variations in single pulse energy and peak power with the absorbed pump power. At the maximum absorbed pump power of 9.51 W, laser pulses with energies of 5.7 µJ, 7.8 µJ, and 7.6 µJ were generated from T = 1%, 3%, and 5% systems, respectively (see Figure 9a). The increased pulse energy and shortened pulse duration make the pulse peak power increase with the absorbed pump power (see Figure 9b). The maximum pulse peak power approaching 38 W was realized at the OC of T = 3%. 

Table 1 summarizes some typical performance of the 3 μm passively Q-switched Er:Lu_2_O_3_ crystal lasers. It shows that the TiB_2_-based Q-switched Er:Lu_2_O_3_ laser has many advantages, such as high output power, relatively high pulse energy, shorter pulse duration, and high pulse peak power, which are attributed to the high laser damage threshold of TiB_2_ SA.

It is possible to achieve shorter laser pulses in TiB_2_-based Q-switched Er:Lu_2_O_3_ crystal lasers. This is because the shortest pulse duration from a passively Q-switched bulk laser is proportional to the laser cavity length while inversely proportional to the SA’s modulation depth [27]. To achieve this, a more compact laser cavity and optimized TiB2 nanoparticles with a larger modulation depth can be used. Further power scaling can also be expected by improving the mode matching between the pumping laser and the oscillated laser inside the laser cavity.

## 6. Conclusions

In this paper, TiB_2_ nanoparticles were prepared via the liquid phase exfoliation method, and their morphology was analyzed using SEM, TEM, and AFM. The TiB_2_ nanoparticles demonstrated a saturable absorption property at 2.85 μm, which was both theoretically predicted and experimentally verified. Employed as an SA, a 2.85 μm pulsed Er:Lu_2_O_3_ laser with an average output power of 1.2 W was achieved under a maximum pump power of 9.51 W. The related shortest pulse duration and pulse energy of 203 ns and 7.8 µJ were delivered at a 154 kHz repetition rate. As far as we know, the reported result represents the highest output power of the Q-switched Er:Lu_2_O_3_ crystal laser. The findings indicate that TiB2, as an SA, possesses several advantages, such as exceptional optical performance, good mechanical and thermal stability, and easy preparation. These outcomes also offer valuable insights into the advancement of pulsed mid-infrared solid-state lasers. By using better materials and improving the quality of the SAs, it is possible to produce a pulsed Er:Lu_2_O_3_ crystal laser with an output power exceeding 2 W.

## Figures and Tables

**Figure 1 nanomaterials-14-00379-f001:**
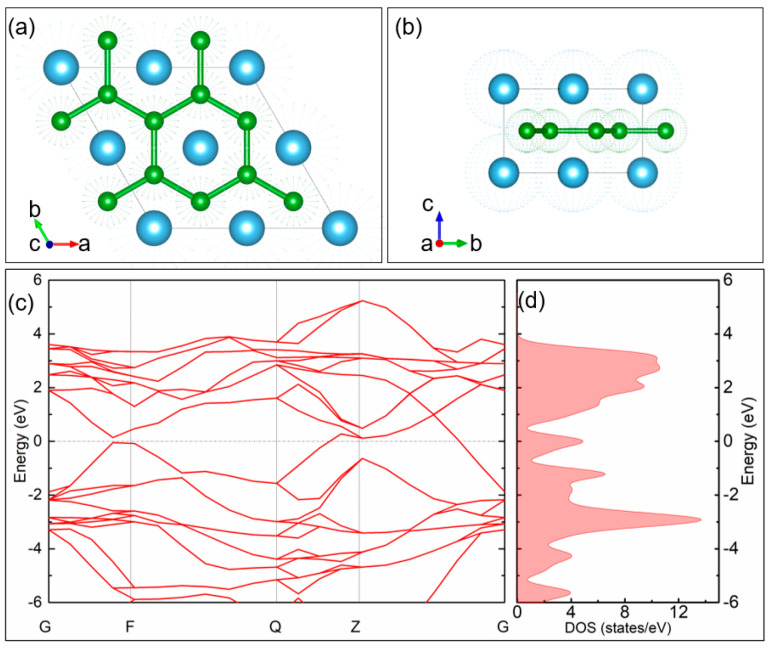
Schematic diagram of the TiB_2_ supercell views down (**a**) c-axis and (**b**) a-axis; the corresponding (**c**) band structure, and (**d**) density of state.

**Figure 2 nanomaterials-14-00379-f002:**
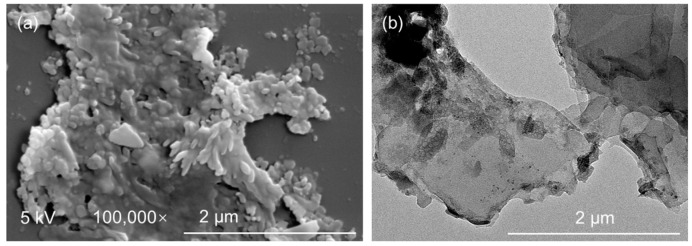
(**a**) SEM and (**b**) TEM image of the prepared TiB_2_ nanoparticles.

**Figure 3 nanomaterials-14-00379-f003:**
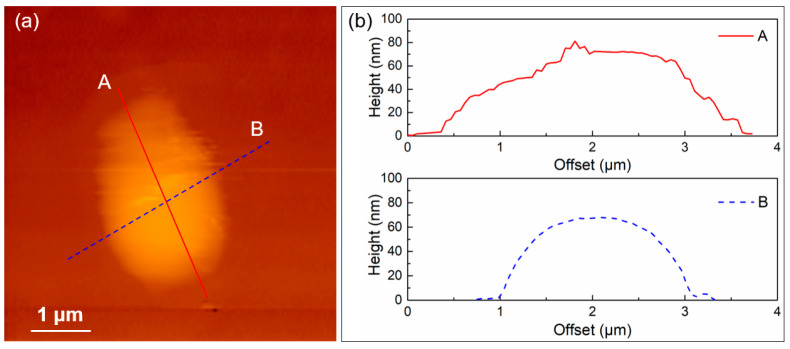
(**a**) AFM image of the TiB_2_ nanoparticles, (**b**) the height variation of TiB_2_ nanoparticles along the section A and B in figure (**a**).

**Figure 4 nanomaterials-14-00379-f004:**
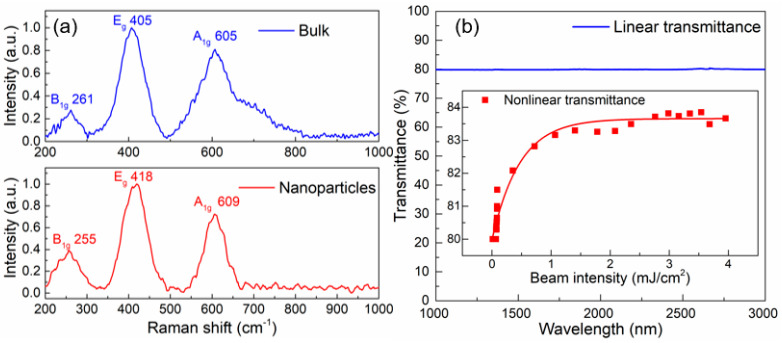
(**a**) Raman spectra of TiB_2_ bulk (**top**) and nanoparticles samples (**bottom**); (**b**) optical transmittance of YAG-based TiB_2_ nanoparticles.

**Figure 5 nanomaterials-14-00379-f005:**
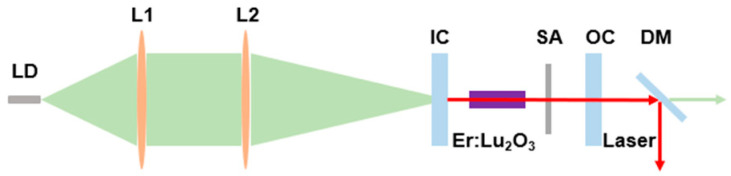
Schematic diagram of the TiB_2_ Q-switched Er:Lu_2_O_3_ laser.

**Figure 6 nanomaterials-14-00379-f006:**
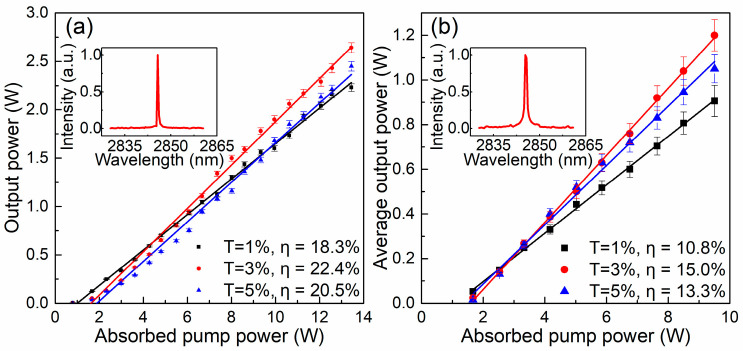
(**a**) Output power of CW Er:Lu_2_O_3_ laser versus absorbed pump power, inset: corresponding lasing spectrum at maximum output power; (**b**) average output power of the Q-switched Er:Lu_2_O_3_ laser versus absorbed pump power, inset: corresponding lasing spectrum at maximum output power.

**Figure 7 nanomaterials-14-00379-f007:**
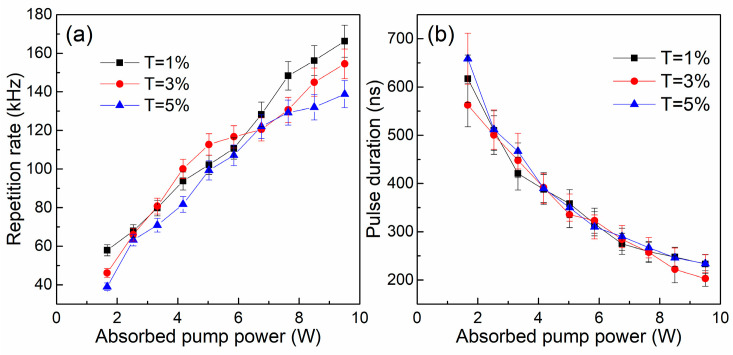
(**a**) pulse repetition rates and (**b**) pulse durations of the Q-switched laser versus absorbed pump power.

**Figure 8 nanomaterials-14-00379-f008:**
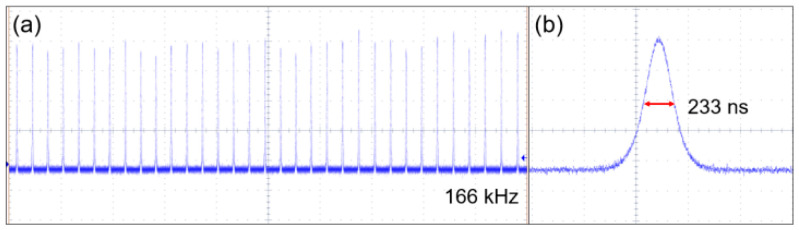
(**a**) Q-switched pulse train and (**b**) single pulse shape.

**Figure 9 nanomaterials-14-00379-f009:**
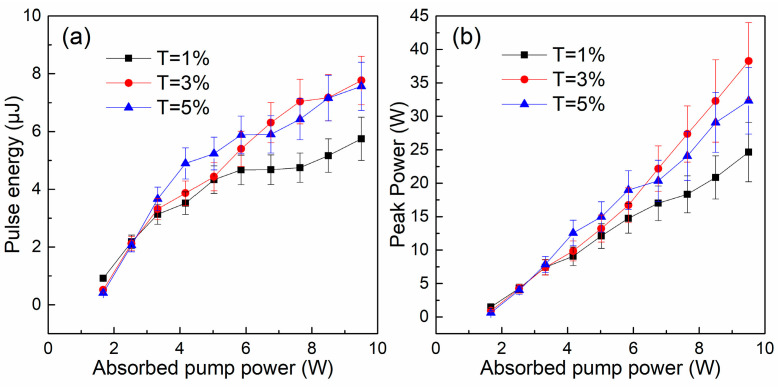
(**a**) pulse energy properties and (**b**) peak power properties of the Q-switched Er:Lu_2_O_3_ laser.

**Table 1 nanomaterials-14-00379-t001:** Summary of performances of the reported passively Q-switched Er:Lu_2_O_3_ crystal lasers.

SA	Average Power (W)	Pulse Energy (μJ)	Pulse Duration (ns)	Peak Power (W)	Ref.
MoS_2_	1.030	8.5	335	25	[25]
BP	0.755	7.1	359	20	[26]
ZrC	0.262	22	50	440	[12]
MXenes Ti_3_C_2_T_x_	1.130	7.6	187	40	[13]
TiB_2_	1.20	7.8	203	39.3	This work

## Data Availability

Data underlying the results presented in this paper are not publicly available at this time but may be obtained from the authors upon reasonable request.

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
