# Peer review of "A Watt-Level Pulsed Er:Lu2O3 Laser Based on a TiB2 Saturable Absorber"

_nanomaterials, 2024, doi:10.3390/nano14040379_

Round 1
Reviewer 1 Report
Comments and Suggestions for Authors
The paper reports on the fabrication of TiB2 nanosheets and their use as saturable absorbers for the pulsed operation of an Er:Lu2O3 laser in the 3 um range. In particular, the fabrication process is briefly discussed, morphology of the nanosheets analyzed by electron (and atomic force) microscopy, their optical and structural properties investigated by absorption and Raman spectroscopy, and their use assessed in a Q-switched laser configuration. Moreover, the band structure of TiB2 nanosheets is calculated, with the main aim to provide a support to experimental results.
The manuscript suffers from style and language issues, sometimes preventing a complete understanding of the reported statements. In particular, choice of terms is often not adequate for a scientific publication.
Assuming that such issues will be fixed prior to publication, my overall evaluation of the scientific content, as far as I could understand due to language limitations, is positive. Despite some excess of conciseness, the paper is rather complete in terms of experimental characterizations and results are convincing, although novelty of the presented problems and related methods is questionable.
Further to the already mentioned problems with style and language, to be definitely fixed, there are a few additional points the Authors are invited to consider, as follows, prior to acceptance.
1. Statement at lines 77-79 must be revised. There, Authors say that, because of a wavelength-independent absorption, TiB2 nanosheets can be retained as a suitable saturable absorber in a 3 um laser. I can’t see the relationship, and I don’t think that the property pointed out is sufficient for a material to be used as an efficient saturable absorber. Authors must better discuss in the text.
2. AFM imaging is used to support the sheet-like morphology of the material. From the image reported in Fig. 3(a) and the cross-sectional profile in panels (b) [and (c)], I could not see a clear evidence of the sheet-like nature. In particular, how large is the minimum sheet height as deduced by AFM? Is it in agreement with expectations? Authors must discuss this point more carefully, and clearly, in the text.
3. Statement at lines 108-111 should also be revised. First of all, it is unclear whether the differences between Raman spectra in the bulk and the sheets are expected, or not, in particular why the intensity distribution and peak location should change. This must be discussed in the text and supported by suitable references. Moreover, that the Raman spectra can prove the successful preparation of nanosheets, as reported in the text, is questionable, in my opinion, unless a quantitative agreement between the experimental Raman spectra and theoretical predictions is given.
4. I can’t see any error bar in plots show in Figs. 6 and 7. Authors must discuss uncertainty of their measurements.
5. Conclusions sound rather weak, and they should be improved, for instance by making reference to the comparison of the results summarized in Table 1.
Comments on the Quality of English Language
Style and language must be deeply revised prior to acceptance. In particular, choice of terms is often not adequate for a scientific publication. Revision by a reader proficient in English and in technical topics (relating with optics, lasers, materials) is highly recommended.
Author Response
Comments and Suggestions for Authors:
The paper reports on the fabrication of TiB2 nanosheets and their use as saturable absorbers for the pulsed operation of an Er:Lu2O3 laser in the 3 um range. In particular, the fabrication process is briefly discussed, morphology of the nanosheets analyzed by electron (and atomic force) microscopy, their optical and structural properties investigated by absorption and Raman spectroscopy, and their use assessed in a Q-switched laser configuration. Moreover, the band structure of TiB2 nanosheets is calculated, with the main aim to provide a support to experimental results.
The manuscript suffers from style and language issues, sometimes preventing a complete understanding of the reported statements. In particular, choice of terms is often not adequate for a scientific publication.
Assuming that such issues will be fixed prior to publication, my overall evaluation of the scientific content, as far as I could understand due to language limitations, is positive. Despite some excess of conciseness, the paper is rather complete in terms of experimental characterizations and results are convincing, although novelty of the presented problems and related methods is questionable.
Further to the already mentioned problems with style and language, to be definitely fixed, there are a few additional points the Authors are invited to consider, as follows, prior to acceptance.
Response: We thank the reviewer for valuable suggestions for improving the quality of the manuscript.
During the revision, we revised the manuscript word by word, which will not change the framework and main content of the article. As too many revisions were performed, to highlight the response to the major suggestions, we did not mark the revision of the language.
We believe that the revised manuscript is now clearer and more readable, making it easier for reviewers and readers to understand its main content.
- Statement at lines 77-79 must be revised. There, Authors say that, because of a wavelength-independent absorption, TiB2 nanosheets can be retained as a suitable saturable absorber in a 3 um laser. I can’t see the relationship, and I don’t think that the property pointed out is sufficient for a material to be used as an efficient saturable absorber. Authors must better discuss in the text.
Response: We thank the reviewer for the valuable suggestions for the manuscript. we revised the statement in the manuscript and marked it in red, as below:
The graphene-liked band structure of the materials allows absorption for any incidence photon [16]. As a tiny sample possesses a certain number of electrons, when a high-brightness light incident on the materials, electrons will be gradually excited to a high energy level until the sample is bleached. In macroscopic terms, the material will show a saturable absorption property. Thus, at ~3 μm, the saturable absorption property of TiB2 is also worth to be expected.
- Dawlaty M. J.; Shivaraman S.; Strait J.; George P.; Chandrashekhar M.; Rana F.; Spencer G. M.; Veksler D.; Chen Y. Meas-urement of the optical absorption spectra of epitaxial graphene from terahertz to visible. Appl. Phys. Lett. 2008, 93, 131905.
- AFM imaging is used to support the sheet-like morphology of the material. From the image reported in Fig. 3(a) and the cross-sectional profile in panels (b) [and (c)], I could not see a clear evidence of the sheet-like nature. In particular, how large is the minimum sheet height as deduced by AFM? Is it in agreement with expectations? Authors must discuss this point more carefully, and clearly, in the text.
Response: We thank the common of the reviewer.`
During the revision, we analyzed the characterization results of the sample, carefully. Even from the TEM image, the sample shows a clear layered structure, the SEM and AFM images remind us the sample may be more of an agglomerated nanoparticle than a nanosheet. Thus, the word “nanosheet” lacks of rigorous.
During the revision, we replaced “nanosheet” with “agglomerated nanoparticles”, and changed the related descriptions in the main text. We believe the correction will improve the quality of the manuscript and eliminate the misleading to the readers.
We thank the reviewer again and hope the revised version will meet the requirements of the journal.
- Statement at lines 108-111 should also be revised. First of all, it is unclear whether the differences between Raman spectra in the bulk and the sheets are expected, or not, in particular why the intensity distribution and peak location should change. This must be discussed in the text and supported by suitable references. Moreover, that the Raman spectra can prove the successful preparation of nanosheets, as reported in the text, is questionable, in my opinion, unless a quantitative agreement between the experimental Raman spectra and theoretical predictions is given.
Response: We thank the reviewer for the valuable suggestions.
The reviewer gives very valuable comments on the Raman spectrum characterization.
The reference of Ľ. Bača, N. Stelzer, Adapting of sol–gel process for preparation of TiB2 powder from low-cost precursors, Journal of the European Ceramic Society, Volume 28, Issue 5, 2008, Pages 907-911 reported the Raman peaks of the TiB2 powder are 262, 404, and 598 cm−1. And reference Liu, K., Feng, J., Guo, J., Chen, L., Feng, Y., Tang, Y., Lu, H., Yu, J., Zhang, J., Zhao, H. and He, T., (1‐10) Facet‐Dominated TiB2 Nanosheets with High Exposure of Dual‐Atom‐Sites for Enhanced Polysulfide Conversion in Li‐S Batteries. Advanced Functional Materials, p.2314657 reported the three peaks of the TiB2 nanosheets are centered at 254, 436, and 609 cm−1, which could be assigned to the three dominant vibration modes (B1g, Eg, and A1g) of TiB2. Comparing the reported results, we can find that Raman peaks are slightly different between TiB2 powders and nanosheets. As the presented Raman characterization is different from the reported results, we agree with the reviewer that the Raman spectrum cannot prove the successful preparation of the nanoparticles. And the intensity distribution may result from the testing.
To clarify the above issues, we retested the Raman spectrum of the TiB2 powders and nanoparticles, during the revision. The result was drawn in Fig. 4(a) of the revised manuscript. The corresponding shift of the peak and individual peak assignment has been properly assigned, as below:
Fig. 4 (a) shows the Raman spectra of both the bulk TiB2 and the TiB2 nanoparticles. Compared with the bulk samples, the three Raman peaks of the TiB2 nanoparticles shift obviously. Detailly, the peak shifts from 261 to 255 cm-1, 261 to 255 cm-1 and 261 to 255 cm-1, corresponding to the vibration modes of B1g, Eg and A1g. If we ignore the tiny error during the measurement, the results meet well with the publications [].
Figure 1. Raman spectra of TiB2 bulk (top) and nanoparticles samples (bottom)
Reference
Bača Ľ.; Stelzer N.; Adapting of sol–gel process for preparation of TiB2 powder from low-cost precursors, J. Eur. Ceram. Soc. 2008, 28, 907-911
Liu K.; Feng J.; Guo J.; Chen L.; Feng Y.; Tang Y.; Lu H.; Yu J.; Zhang J.; Zhao H.; He, T. (1-10) Facet-Dominated TiB2 Nanosheets with High Exposure of Dual‐Atom‐Sites for Enhanced Polysulfide Conversion in Li-S Batteries. Adv. Funct. Mater. 2023. 2314657
- I can’t see any error bar in plots show in Figs. 6 and 7. Authors must discuss uncertainty of their measurements.
Response: Thanks for the common on the uncertainty of their measurements.
In the previous version, we only displayed the average values of the output powers, repetition rates, and pulse durations. However, in the revised manuscript, we have included error bars in Figs. 6 and 7 to demonstrate the uncertainty of the measurements. Additionally, the uncertainties of the pulse energy and peak power, which result from the output powers, repetition rates, and pulse durations, have also been shown in Fig. 9 with error bars.
The related description in the text has also been revised and marked in red.
- Conclusions sound rather weak, and they should be improved, for instance by making reference to the comparison of the results summarized in Table 1.
Response: We thank the valuable common, we revised the conclusion and marked it in red in the manuscript, as below:
In this paper, TiB2 nanoparticles was prepared by the liquid phase exfoliation method, and the morphology was systematically characterized by the SEM, TEM and AFM. The saturable absorption property of the TiB2 nanoparticles at 2.85 μm was theoretically predicted and experimentally proved. Employed as an SA, a 2.85-μm pulsed Er:Lu2O3 laser with an average output power of 1.2 W was achieved under a maximum pump power of 9.51 W. The related shortest pulse duration and pulse energy of 203 ns and 7.8 µJ were delivered at a 154 kHz repetition rate. As far as we know, the reported result represents the highest output power of the Q-switched Er:Lu2O3 crystal laser. The findings indicate that TiB2, as an SA, possesses several advantages such as exceptional optical performance, good mechanical and thermal stability, and easy preparation. These outcomes also offer valuable insights into the advancement of pulsed mid-infrared solid-state lasers. By further selecting better materials and improving the quality of the SAs, a pulsed Er:Lu2O3 crystal laser with output power over 2 W is worth to be expected.
Comments on the Quality of English Language
Style and language must be deeply revised prior to acceptance. In particular, choice of terms is often not adequate for a scientific publication. Revision by a reader proficient in English and in technical topics (relating with optics, lasers, materials) is highly recommended.
Response: We thank the reviewer for the valuable suggestions on the language.
During the revision, we revised the manuscript word by word, which will not change the framework and main content of the article. As too many revisions were performed, to highlight the response to the major suggestions, we did not mark the revision of the language.
We hope the revied manuscript will meet the requirement of the reviewer.

Reviewer 2 Report
Comments and Suggestions for Authors
The paper was well written. I have only one minor comments.
1. The brief explanation of aturable absorbers should be added. It would help readers.
Author Response
The paper was well written. I have only one minor comments.
- The brief explanation of saturable absorbers should be added. It would help readers.
Response: We thank the reviewer for the valuable suggestions for the manuscript. we made revisions to state a brief explanation of the mechanism of the saturable absorption property of the TiB2 sample in the manuscript and marked it in red, as below:
The graphene-liked band structure of the materials allows absorption for any incidence photon [16]. As a tiny sample possesses a certain number of electrons, when a high-brightness light incident on the materials, electrons will be gradually excited to a high energy level until the sample is bleached. In macroscopic terms, the material will show a saturable absorp-tion property. Thus, at ~3 μm, the saturable absorption property of TiB2 is also worth to be expected.
- Dawlaty M. J.; Shivaraman S.; Strait J.; George P.; Chandrashekhar M.; Rana F.; Spencer G. M.; Veksler D.; Chen Y. Meas-urement of the optical absorption spectra of epitaxial graphene from terahertz to visible. Appl. Phys. Lett. 2008, 93, 131905.
We noticed that there are some tiny mistakes in the description of the characterization of the TiB2 sample. and the revision was performed and marked in red in the text,during the revision.
Reviewer 3 Report
Comments and Suggestions for Authors
Liang et al. have demonstrated nanostructure TiB2 saturable absorber for Watt-level pulsed Er:Lu2O3 laser . They claim the nanostructure TiB2 are nanosheet and are prepared via solution based methods. They further used it as saturable absorber in a Er:Lu2O3 crystal laser. My comments are outlined below
1. The author claim the TiB2 nanostructure are nanosheet. However, there is not enough evidence provided to support the claim. From SEM images they more look like agglomerated nanoparticles. Similarly, the author show different region of SEM, TEM and AFM images. The evidence provided make it’s really hard to conclude indeed they are nanosheet. Their claim on AFM scans doesn’t provide solid evidence. Hence, my suggestion is not to use the word nanosheet in the manuscript.
2. The Raman spectra provided for TiB2 nanosheet doesn’t agree with previous report. The shift of the peak and individual peak assignment should be properly assigned. For example the author should compare their work with “Liu, K., Feng, J., Guo, J., Chen, L., Feng, Y., Tang, Y., Lu, H., Yu, J., Zhang, J., Zhao, H. and He, T., (1‐10) Facet‐Dominated TiB2 Nanosheets with High Exposure of Dual‐Atom‐Sites for Enhanced Polysulfide Conversion in Li‐S Batteries. Advanced Functional Materials, p.2314657”
3. In figure9, the plot between pulse energy and absorbed pulse power, there is plateau for certain region of absorbed pump power. The author should explain it.
4. There are some grammatical mistake, the author should thoroughly review it.
Comments on the Quality of English Language
Minor edits are required.
Author Response
Liang et al. have demonstrated nanostructure TiB2 saturable absorber for Watt-level pulsed Er:Lu2O3 laser . They claim the nanostructure TiB2 are nanosheet and are prepared via solution based methods. They further used it as saturable absorber in a Er:Lu2O3 crystal laser. My comments are outlined below
- The author claim the TiB2nanostructure are nanosheet. However, there is not enough evidence provided to support the claim. From SEM images they more look like agglomerated nanoparticles. Similarly, the author show different region of SEM, TEM and AFM images. The evidence provided make it’s really hard to conclude indeed they are nanosheet. Their claim on AFM scans doesn’t provide solid evidence. Hence, my suggestion is not to use the word nanosheet in the manuscript.
Response: We thank the kind reminder from reviewer.
We agree with the reviewer, that the illustrated results of SEM, TEM and AFM images show the prepared sample is more of a cluster than a nanosheet. During the revision, we replaced “nanosheet” with “agglomerated nanoparticles”, and changed some descriptions in the manuscript. We believe the revision will eliminate the mistakes in the previous version.
Thanks again.
- The Raman spectra provided for TiB2nanosheet doesn’t agree with previous report. The shift of the peak and individual peak assignment should be properly assigned. For example the author should compare their work with “Liu, K., Feng, J., Guo, J., Chen, L., Feng, Y., Tang, Y., Lu, H., Yu, J., Zhang, J., Zhao, H. and He, T., (1‐10) Facet‐Dominated TiB2 Nanosheets with High Exposure of Dual‐Atom‐Sites for Enhanced Polysulfide Conversion in Li‐S Batteries. Advanced Functional Materials, p.2314657”
Response: We thank the reviewer for the comment.
Reference Liu, K., Feng, J., Guo, J., Chen, L., Feng, Y., Tang, Y., Lu, H., Yu, J., Zhang, J., Zhao, H. and He, T., (1‐10) Facet‐Dominated TiB2 Nanosheets with High Exposure of Dual‐Atom‐Sites for Enhanced Polysulfide Conversion in Li‐S Batteries. Advanced Functional Materials, p.2314657 reported the three peaks of the TiB2 nanosheets are centered at 254, 436, and 609 cm−1 (B1g, Eg, and A1g), and the mass bulk samples, the Raman peaks are 262, 404, and 598 cm−1 (Ľ. Bača, N. Stelzer, Adapting of sol–gel process for preparation of TiB2 powder from low-cost precursors, Journal of the European Ceramic Society, Volume 28, Issue 5, 2008, Pages 907-911). our characterization is slightly different from the results of the above publications.
During the revision, we retested the Raman spectrum of the TiB2 powders and nanoparticles and drew it in Fig. 4(a) of the revised manuscript. The corresponding shift of the peak and individual peak assignment has been properly assigned, as below:
Figure 1. Raman spectra of TiB2 bulk (top) and nanoparticles samples (bottom)
- In figure9, the plot between pulse energy and absorbed pulse power, there is plateau for certain region of absorbed pump power. The author should explain it.
Response: Thanks for the common, we would like to respond to the common here:
Usually, the sample prepared by the spin-coated method lacks uniformity, and it is difficult to ensure that the same part of the sample is used in different experiments. Thus, the lasing conditions are slightly different in the three Q-switched performances, as a result, the pulse energy curves vary in different rules with increasing of the pump power, in three lasers.
In our experiment, the plateau emerges in all three curves (T = 1% @ 6~8 W, T = 3% @ 8-9 W and T = 5% @ 6-7 W). We believe it is resulting from the saturation of the active aera of the SA, so the energy will not increase further. However, by further increasing the pump power, the mode field area in the laser cavity increased. the active area of SA will also increase, simultaneously, which corresponds to a higher saturation threshold and enables a higher laser energy.
- There are some grammatical mistake, the author should thoroughly review it.
Response: We thank the reviewer for the valuable suggestions.
During the revision, we revised the manuscript word by word, which will not change the framework and main content of the article. As too many revisions were performed, to highlight the response to the major suggestions, we did not mark the revision of the language.
Round 2
Reviewer 3 Report
Comments and Suggestions for Authors
The author answered all my queries and re-modified the manuscript.
Comments on the Quality of English Language
Minor changes are needed.